# Clinical Features and Vaccination Effects among Children with Post-Acute Sequelae of COVID-19 in Taiwan

**DOI:** 10.3390/vaccines12080910

**Published:** 2024-08-12

**Authors:** Yu-Lung Hsu, Pei-Chi Chen, Yi-Fen Tsai, Chi-Hung Wei, Lawrence Shi-Hsin Wu, Kai-Sheng Hsieh, Miao-Hsi Hsieh, Huan-Cheng Lai, Chien-Heng Lin, Hsiao-Chuan Lin, Chieh-Ho Chen, An-Chyi Chen, Hung-Chih Lin, I-Ching Chou, Wen-Jue Soong, Kao-Pin Hwang, Henry Horng-Shing Lu, Ruby Pawankar, Hui-Ju Tsai, Jiu-Yao Wang

**Affiliations:** 1Division of Pediatric Infectious Diseases, China Medical University Children’s Hospital, China Medical University, Taichung 40447, Taiwan; codan5230@gmail.com (Y.-L.H.); 018420@tool.caaumed.org.tw (H.-C.L.); 004267@tool.caaumed.org.tw (H.-C.L.); 008680@tool.caaumed.org.tw (K.-P.H.); 2Graduate Institute of Biomedical Sciences, China Medical University, Taichung 40447, Taiwan; lshwu@hotmail.com; 3Center of Allergy, Immunology and Microbiome (A.I.M), China Medical University Hospital, China Medical University Children’s Hospital, Taichung 40447, Taiwan; simple48686@gmail.com (P.-C.C.); karinadrift@gmail.com (M.-H.H.); 4Department of Microbiology & Immunology, College of Medicine, National Cheng Kung University, Tainan 70101, Taiwan; 5Institute of Population Health Sciences, National Health Research Institutes, Zhunan 350401, Taiwan; puda@nhri.edu.tw; 6Division of Medical Research, China Medical University Children’s Hospital, China Medical University, Taichung 40447, Taiwan; 081673@tool.caaumed.org.tw (C.-H.W.); 008893@tool.caaumed.org.tw (K.-S.H.); 7Division of Pediatric Pulmonology and Intensive Care, China Medical University Children’s Hospital, China Medical University, Taichung 40447, Taiwan; 010716@tool.caaumed.org.tw (C.-H.L.); 030270@tool.caaumed.org.tw (C.-H.C.); 008963@tool.caaumed.org.tw (W.-J.S.); 8Division of Pediatric Gastroenterology, China Medical University Children’s Hospital, China Medical University, Taichung 40447, Taiwan; 008427@tool.caaumed.org.tw; 9Division of Neonatology, China Medical University Children’s Hospital, China Medical University, Taichung 40447, Taiwan; 000373@tool.caaumed.org.tw; 10Division of Pediatric Neurology, China Medical University Children’s Hospital, China Medical University, Taichung 40447, Taiwan; 004009@tool.caaumed.org.tw; 11Institute of Statistics, National Yang Ming Chiao Tung University, Hsinchu 30010, Taiwan; hslu@stat.nycu.edu.tw; 12Department of Pediatrics, Nippon Medical School, Tokyo 8602, Japan; pawankar.ruby@gmail.com; 13Department of Medical Science, National Tsing-Hua University, Hsinchu 300044, Taiwan

**Keywords:** children, COVID-19, SARS-CoV-2, post-acute sequelae of SARS-CoV-2 infection, vaccination

## Abstract

Background: Post-acute sequelae of SARS-CoV-2 infection (PASC) affects patients after recovering from acute coronavirus disease 2019 (COVID-19). This study investigates the impact of SARS-CoV-2 vaccination on PASC symptoms in children in Taiwan during the Omicron pandemic. Methods: We enrolled children under 18 years with PASC symptoms persisting for more than 4 weeks. Data collected included demographics, clinical information, vaccination status, and symptom persistence. We used logistic regression models to compare symptoms in the acute and post-COVID-19 phases and to assess the association between vaccination and these symptoms. Results: Among 500 PASC children, 292 (58.4%) were vaccinated, 282 (52.8%) were male, and the mean (SD) age was 7.6 (4.6) years. Vaccinated individuals exhibited higher odds of experiencing symptoms in the previous acute phase, such as cough (adjusted odds ratio [AOR] = 1.57; 95% confidence interval [CI]: 1.02–2.42), rhinorrhea/nasal congestion (AOR = 1.74; 95% CI: 1.13–2.67), sneezing (AOR = 1.68; 95% CI: 1.02–2.76), sputum production (AOR = 1.91; 95% CI: 1.15–3.19), headache/dizziness (AOR = 1.73; 95% CI: 1.04–2.87), and muscle soreness (AOR = 2.33; 95% CI: 1.13–4.80). In contrast, there were lower odds of experiencing abdominal pain (AOR = 0.49; 95% CI: 0.25–0.94) and diarrhea (AOR = 0.37; 95% CI: 0.17–0.78) in children who had received vaccination during the post-COVID-19 phase. Conclusions: This study revealed clinical features and vaccination effects in PASC children in Taiwan. Vaccination may reduce some gastrointestinal symptoms in the post-COVID-19 phase.

## 1. Introduction

The global coronavirus disease 2019 (COVID-19) pandemic, instigated by the severe acute respiratory syndrome coronavirus 2 (SARS-CoV-2) virus, has resulted in millions of infections and a broad spectrum of clinical manifestations and outcomes [1]. While most people recover from the acute phase of COVID-19 within weeks, a subset of patients experience persistent or new symptoms that last for months after the initial infection [2]. This condition has been termed post-acute sequelae of SARS-CoV-2 infection (PASC) or long COVID, and can affect multiple organ systems and domains of functioning, such as respiratory, cardiovascular, neurological, psychological, and cognitive sequelae [3].

Currently, the epidemiology, pathophysiology, and management of PASC are still inadequately comprehended, especially in children. Although children generally have milder or asymptomatic COVID-19 compared to adults, some children may develop PASC that can impair their health and well-being [4]. The prevalence of PASC in children varies widely, due to their heterogenous nature, with estimated prevalence ranging from 0.02% to 66.5% [4,5,6,7,8]. Symptoms of PASC in children include fatigue, dyspnea, cognitive impairment, muscle pain, headache, fever dyspnea, chest pain, anxiety, depression, and even multisystem inflammatory syndrome [7,8,9,10,11]. Risk factors and predictors of PASC in children include age, sex, comorbid conditions, severe acute COVID-19, and positive antibody test [4,12,13,14]. Potential mechanisms of PASC in children may involve in organ damage, viral persistence, immune dysregulation, microbiome/virome alterations, clotting/coagulation issues, brainstem/vagus nerve dysfunction, primed immune cells, and autoimmunity [15,16,17,18].

One of the key strategies to prevent COVID-19 is vaccination against SARS-CoV-2. Vaccination can reduce the risk of infection, transmission, hospitalization, and death from COVID-19 [19,20,21]. Vaccination may reduce the risk of PASC’s development and/or the severity of PASC by preventing reinfection or enhancing the immune response against SARS-CoV-2 [22,23,24]. Nevertheless, evidence is scant regarding the impact of vaccination on children with PASC, especially in the context of the Omicron pandemic [24,25].

In this study, we investigated clinical features and vaccination effects in Taiwanese children with PASC. We aimed to characterize clinical symptoms during the acute and post-COVID-19 phases, determine the association of clinical symptoms between these two phases, and assess the impact of vaccination on clinical presentations in both phases in children with PASC.

## 2. Methods

### 2.1. Data Source and Study Participants

The data used in the present cross-sectional study are from a prospective cohort that was conducted at the China Medical University Children’s Hospital (CMUCH), a single medical tertiary center in central Taiwan. The study participants were children from the DISCOVER study cohort (Diagnosis and Support for COVID Children to Enhance Recovery). The DISCOVER cohort was a multidisciplinary combined-care program which was initiated to investigate the clinical conditions and to provide supportive management for children with previously verified SARS-CoV-2 infection by reverse transcriptase-polymerase chain reaction (RT-PCR) or rapid antigen test, and PASC conditions lasting longer than 4 weeks [4,26]. Participants in this study were specifically evaluated in the PASC outpatient department to ensure they were experiencing post-COVID-19 conditions and not undergoing any acute infections. This evaluation included an assessment by a pediatrician to exclude any other acute infections or other causes that mimic post-COVID-19 symptoms.

The participants were recruited to the DISCOVER cohort from 1 July 2022, to 31 July 2023, during the Omicron variant outbreak of the SARS-CoV-2 pandemic in Taiwan, which started in April 2022. PASC was defined in children as persisting symptoms for four weeks after recovering from acute COVID-19 infection [4,26]. All methods in this study were performed in accordance with the relevant guidelines and regulations. This study was approved by the Institutional Review Board of China Medical University Hospital (CMUH111-REC2-113 and CMUH111-REC2-122).

For the sample size calculation, we set the significance level (α) at 0.05 and aimed for a power of 80% (β = 0.20). The proportion of subjects in the exposed group was assumed to be 0.58, based on the distribution of children with vaccination observed in our study population. The baseline risk in the non-exposed group was estimated to be 0.25, based on the previous result from a meta-analysis [6], with an odds ratio (OR) of 1.8. Using these parameters, the estimated sample size required was approximately 443 subjects without applying a continuity correction. When continuity correction was ap-plied, the sample size increased to 476 subjects.

### 2.2. Data Collection

A questionnaire was administered to children and parents who visited the outpatient department and sought medical care for post-COVID-19 condition(s). The questionnaire included demographic information, COVID-19 vaccination status, symptoms experienced, management of acute COVID-19 infection, and any persisting symptoms observed four weeks after recovery. Laboratory examinations were conducted, including biomedical tests such as complete blood cell count, differential count, erythrocyte sedimentation rate (ESR), high-sensitivity C-reactive protein (hsCRP), lactic dehydrogenase (LDH), aspartate aminotransferase (AST), alanine transaminase (ALT), creatine phosphokinase (CPK), and immunoglobulin E (IgE). By analyzing biomarkers from the blood samples of a subset of participants, we sought to understand the underlying mechanisms and physiological changes associated with PASC.

### 2.3. Statistical Analysis

Descriptive statistics for demographic and clinical features, including age, sex, vaccination status, and biomedical test results, were reported as counts with corresponding percentages or as means with corresponding standard deviations (SDs). The study participants were divided into two groups based on vaccination status (vaccinated and unvaccinated) and into three age groups (0–5 years, 6–11 years, and 12–17 years). To compare demographic and clinical characteristics across these groups, Student’s t-test was used for continuous variables, and the chi-square test was employed for discrete variables. We generated heatmap plots to present the frequencies of the first ten common clinical symptoms in acute and post-COVID-19 phases, separately, among 500 study children. We performed the McNemar test to compare clinical symptoms between acute and post-COVID-19 phases among the study children (paired data). Logistic regression models with age and sex adjustment were applied to investigate associations of clinical symptoms during acute and post-COVID-19 phases, separately, with vaccination in children with PASC. We estimated statistical uncertainty of associations using 95% confidence intervals (CIs). Statistical significance was declared using a p-value less than 0.05. All plots and analyses were performed using SAS version 9.4 for Windows (SAS Institute, Cary, NC, USA) and the matplotlib and seaborn packages with Python version 3.11.5.

## 3. Results

### 3.1. Baseline Characteristics of the Study Children

A total of 500 children with PASC were included in this study. Table 1 presents the demographic and clinical characteristics of the study participants (282 (56.4%) boys; mean age ± standard deviation: 7.6 ± 4.6 years). Among them, 58.8% tested positive by RT-PCR for SARS-CoV-2, 58.4% were vaccinated, and 7.4% were hospitalized during the acute phase.

Compared to children without vaccination, children with vaccination tended to be of an older age (9.7 ± 4.1 vs. 4.6 ± 3.5 years) and have greater levels of neutrophils (50.33 ± 11.24 vs. 42.78 ± 14.40 percent), hemoglobin (13.09 ± 1.48 vs. 12.72 ± 1.47 g/dL), and total IgE (358.19 ± 640.76 vs. 231.88 ± 351.76 IU/mL), and lower levels of lymphocytes (38.11 ± 10.98 vs. 45.31 ± 15.10 percent), platelets (301.95 ± 71.53 vs. 387.13 ± 123.06 × 10^3^ per μL), lactic dehydrogenase (LDH) (188.90 ± 49.53 vs. 222.88 ± 63.97 U/L), and D-dimer (290.78 ± 156.03 vs. 349.88 ± 223.95 ng/mL) (Table 1). The demographic and clinical characteristics of the children with PASC in the three age groups (0–5; 6–11; and 12–17 years) are shown in Table 2. Vaccination status varied significantly among age groups (*p* < 0.0001). In the 0–5-year group, 77.8% were unvaccinated, while in the 6–11-year group, 43.7% had received one dose, and in the 12–17-year group, 42.1% had received three doses. Neutrophil percentages were significantly higher in the 12–17-year group (56.08 ± 8.73 percent) compared to the 0–5-year group (40.12 ± 15.22 percent) and the 6–11-year group (47.93 ± 10.2 percent) (*p* < 0.0001). Conversely, lymphocyte percentages were significantly higher in the 0–5-year group (47.42 ± 16.19 percent) compared to the 12–17-year group (32.75 ± 9.18 percent) (*p* < 0.0001). LDH levels were significantly higher in the 0–5-year group (242.97 ± 67.13 U/L) compared to the 12–17-year group (153.00 ± 33.90 U/L) (*p* < 0.0001). Ferritin levels differed significantly (*p* = 0.01), being highest in the 12–17-year group (65.15 ± 53.55 ng/mL). D-dimer levels were also significantly different (*p* < 0.0001), with the highest levels in the 0–5-year group (383.49 ± 251.73 ng/mL). Aspartate aminotransferase (AST) levels were significantly higher in the 0–5-year group (30.09 ± 11.60 U/L) compared to the 12–17-year group (17.77 ± 9.95 U/L) (*p* < 0.0001).

### 3.2. Common Clinical Symptoms during Acute and Post-COVID-19 Phases

The first ten common clinical symptoms during the acute and post-COVID-19 phases, respectively, among the 500 children with PASC are summarized in Figure 1 and Figure 2. The first ten common clinical symptoms in the acute phase include fever, cough, rhinorrhea/ nasal obstruction, sore throat, fatigue, headache/ dizziness, sneezing, sputum, chills, and muscle soreness (Figure 1). The first ten common clinical symptoms in the post-COVID-19 phase include cough, fatigue, throat problems, limited daily activity, body weight changes, attention disturbances, lack of motivation, chest pain, and decreased appetite (Figure 2). The complete clinical symptoms during the acute and post-COVID-19 phases among the 500 children with PASC are provided in Appendix A.

When comparing clinical symptoms between the acute and post-COVID-19 phases, we found children with fever, cough, and muscle soreness in the acute phase were related to all of the top ten symptoms in the post-COVID-19 phase (Figure 3; Appendix A). However, different to fever, cough, and muscle soreness in the acute phase, children with sneezing and sputum in the acute phase were only related to fatigue and cough in the post-COVID-19 phase (Figure 3; Appendix A).

### 3.3. Association between Clinical Symptoms in Two Phases and Vaccination

We further performed analyses to evaluate whether clinical symptoms in the acute and post-COVID-19 phases, respectively, were associated with vaccination in children. Figure 4 shows that significant positive associations of clinical symptoms with vaccination were found in the acute phase (cough [adjusted odds ratios (AOR) = 1.57; 95% confidence interval (CI): 1.02–2.42]; rhinorrhea/ nasal congestion [AOR = 1.74; 95% CI: 1.13–2.67]; sneezing [AOR = 1.68; 95% CI: 1.02–2.76]; sputum [AOR = 1.91; 95% CI: 1.15–3.19]; headache/dizziness [AOR = 1.73; 95% CI: 1.04–2.87]; and muscle soreness [AOR = 2.33; 95% CI: 1.13–4.80]). Different to clinical symptoms in the acute phase, the results in Figure 5 are evident of significant reverse associations between abdominal pain ([AOR = 0.49; 95% CI: 0.25–0.94]) and diarrhea ([AOR = 0.37; 95% CI: 0.17–0.78]) in the post-COVID-19 phase and vaccination.

## 4. Discussion

In this study, we described the clinical features and investigated the effects of vaccination in children with PASC within the DISCOVER cohort from Taiwan. We found that children with PASC had various symptoms, such as fever, chills, respiratory symptoms, headache/dizziness fatigue, and muscle soreness, in the acute phase, whereas they experienced fatigue, limited daily activity, respiratory discomfort, decreased appetite, body weight changes, attention disturbances, and a lack of motivation in the post-COVID-19 phase. Vaccination may have a protective effect on abdominal pain and diarrhea during the post-COVID-19 phase, but not in the acute phase during COVID-19 infection. Our findings have important implications for understanding the association of clinical symptoms between the acute and post-COVID-19 phases as well as the effects of vaccination in these two phases, which can provide clinicians with greater awareness and management guidance for children with PASC.

Previous studies have reported that PASC affects 0.02% to 66.5% of pediatric COVID-19 cases [4,5,6]. The most frequently reported symptoms of PASC in children are fatigue, exercise intolerance, respiratory problems, and cognitive impairment [4,5,6]. These reported symptoms are similar to those observed in our study, suggesting that PASC may have consistent clinical presentation across different pediatric populations. However, other symptoms of PASC in children, such as headache, gastrointestinal problems, sleep disturbances, and mood changes [4,5,6], were less common in our study. The differences in PASC symptoms may reflect variability due to demographic, genetic, environmental, and social factors, etc.

We also explored the differences in PASC symptoms by three age groups. The observed age differences in symptom manifestations are consistent with previous studies [5,12,27,28,29], which may be due to several factors. First, younger children may have more immature immune systems and lower expression of angiotensin-converting enzyme 2, the receptor for SARS-CoV-2 entering cells, resulting in less severe respiratory involvement and less systemic inflammation [30]. Secondly, younger children may have been less exposed to other coronaviruses or allergens, leading to less cross-reactivity or hypersensitivity of the respiratory mucosa [31]. Further studies are needed to elucidate the underlying mechanisms of age differences in children with PASC.

We found children with acute COVID-19 symptoms, such as fever, cough, rhinorrhea/nasal congestion, sore throat, and muscle soreness, are significantly related to multiple symptoms of PASC (Figure 3). Previous studies have reported that some acute COVID-19 symptoms, for example, cough, dyspnea, and fatigue, were associated with increased risk of PASC symptoms in adults [32,33]. However, these findings were seldom reported in pediatric populations. Most studies were focused on the association between the severity of acute COVID-19 infection and the risk of PASC [34,35,36]. In a previous pediatric study conducted in Italy, Morello and colleagues revealed that certain acute COVID-19 symptoms, including chest pain, dyspnea at rest, dyspnea during exercise, as well as the loss or alteration of smell and taste, exhibited the strongest associations with the persistence of symptoms [37]. Our findings might also infer that some symptoms in the acute phase might be potential indicators for Asian children developing PASC.

In a previous cohort study [4] and review [38], it was indicated that myocarditis and transient or persistent cardiac complications could be induced after COVID-19. Post-COVID-19 symptoms such as fatigue, chest pain, palpitations, and difficulty breathing may be indicative of myocarditis or other cardiac complications. However, myocarditis was not diagnosed among the participants in our study. Additionally, another study conducted by our team, which involved reviewing cardiac ultrasound data, also found no evidence of myocarditis in our PASC cohort. These findings suggest that while symptoms indicative of myocarditis were observed, there was no clinical diagnosis of myocarditis in our cohort. This discrepancy highlights the complexity of PASC.

Elevated plasma lactate dehydrogenase (LDH) levels have been linked to post-acute sequelae of SARS-CoV-2 infection in adults [39], but this association has not been documented in the pediatric population. In our study, we observed that plasma LDH levels in the pediatric cohort remained within the normal range, although there was a notable increase in the unvaccinated group (Table 1). In adults who have survived COVID-19 and continue to experience persistent symptoms, there is a notable trend toward elevated levels of neutrophils, neutrophil-to-lymphocyte ratio (NLR), fibrinogen, D-dimer, ferritin, and C-reactive protein (CRP), coupled with decreased lymphocyte counts. These markers may suggest an uncontrolled inflammatory response as a potential contributor to the development of PASC [40,41]. In a pediatric study, similar patterns were observed, with increased leukocytes, monocytes, neutrophils, platelets, and D-dimer emerging as strong laboratory predictors for PASC [42]. However, in our study, we did not observe elevations in these parameters. Osmanov and colleagues have suggested that allergic diseases might be considered as potential risk factors for PASC [14]. Notably, the levels of IgE in our study population were found to be higher than the normal range (<100 IU/mL), which is consistent with the observation that children with allergic conditions may be more predisposed to developing PASC [14].

The effect of COVID-19 vaccination on PASC symptoms in children is still unclear. There is limited evidence on the efficacy of COVID-19 vaccines in children with PASC [43], and the effect of vaccination on PASC is inconclusive [44,45], particularly following infection with the Omicron variant [46]. Potential benefits of vaccination for PASC symptoms encompass several mechanisms, including the clearance of residual viral particles, an enhancement of immune regulation, the modulation of inflammation, and a restoration of vascular and neurological function [47]. In contrast, vaccination may also exacerbate some PASC symptoms through pathways such as autoimmunity induction, elevation of inflammation, adverse events, and interference with epigenetic mechanisms [48,49,50]. As such, the effect of vaccination on PASC symptoms appears to be intricately linked to patients’ biological factors, including genetic predisposition, immune status, viral load, and any underlying comorbidities [15]. In our study, we found that vaccination was associated with lower odds of abdominal pain and diarrhea in the post-COVID-19 phase, which may indicate a protective effect of vaccination on the gastrointestinal system (Figure 4). One possible explanation for this protective effect is that vaccination may enhance the immune response, reducing viral load and mitigating the inflammatory response in the gastrointestinal tract. It is also important to consider the possibility that this finding could be coincidental. The gastrointestinal symptoms observed might have been influenced by other factors not accounted for in our study, such as differences in diet or underlying health conditions. Otherwise, we also found that vaccination was associated with higher odds of cough, rhinorrhea/nasal congestion, sneezing, sputum, headache/dizziness, and muscle soreness in the acute phase, which may indicate a detrimental effect of vaccination on the respiratory system (Appendix A). These findings are consistent with some reports of different reactions in the acute phase and the post-COVID-19 phase according to vaccination status [51,52,53]. It is possible that vaccination may trigger an immune response that exacerbates or prolongs symptoms of acute COVID-19 infection in children. These significant positive associations between vaccination and certain acute phase symptoms in children with PASC may seem counterintuitive since vaccination is generally expected to provide protection. However, it is important to note that our study population consisted solely of children who developed PASC, which limits our ability to assess the broader protective effect of vaccination. Our results reflect the symptom profiles of vaccinated versus unvaccinated children within this specific group of PASC patients, rather than the general population. Additionally, we took steps to ensure that the symptoms were not caused by other ongoing infections, further supporting the specificity of our findings to PASC-related symptoms. Further studies are needed to confirm the relationship between vaccination and clinical symptoms in children with PASC, and to elucidate the underlying mechanisms.

This study has several strengths. First, this study employs a prospective design to collect data in a real-time fashion, thereby improving the reliability of our findings compared to the retrospective design conducted in most previous studies. Secondly, this study features a predominant representation of children infected with the Omicron variant. This enables us to explore the specific impacts of the Omicron variant on children with PASC. Third, while most studies reported clinical features and vaccination effects in Caucasian pediatric populations [7,8,54], this is one of the few studies reporting clinical features and vaccination effect in Asian children with PASC. Fourth, to the best of our knowledge, this is one of the largest Asian pediatric cohorts evaluating clinical features and vaccination effects related to PASC.

However, this study exhibits certain limitations that warrant consideration. First, the lack of available data on the prevalence and incidence rates of PASC in children in Taiwan during the study period. Due to this, we could not estimate the sample size based on these specific epidemiologic parameters. Instead, our sample size was determined by practical considerations, including available resources, time constraints, and our capacity to recruit and follow up with participants. This approach was intended to strike a balance between scientific rigor and logistical feasibility, but it may affect the generalizability of our findings. Second, it relies on self-reported symptoms, which could be susceptible to recall bias, potentially affecting the accuracy of our results. Third, we did not undertake a comparative evaluation of the effects of different vaccine types in children with PASC. It will be of interest to investigate vaccine efficacy and safety profiles across various kinds of vaccines for PASC prevention. Fourth, future investigations should also explore the duration of protection across various kinds of vaccines and identify who might benefit most from booster shots. These are crucial for developing an optimal vaccination strategy for children with PASC. Fifth, unmeasured confounding effects remain a concern because we were not able to account for certain potential confounders, such as genetic, environmental, or behavioral factors. Thus, the results should be interpreted with caution. Sixth, the data were collected at the first visit to our PASC outpatient department, which might not fully capture the long-term consequences of PASC in children.

## 5. Conclusions

In conclusion, this study provides a comprehensive investigation of baseline characteristics and clinical symptoms in both the acute and post-COVID-19 phases, and links vaccination effects to both the acute and post-COVID-19 phases in Asian children with PASC. Vaccination may reduce certain clinical gastrointestinal symptoms in the post-COVID-19 phase among children with PASC. Further studies are needed to confirm our findings in different populations and to elucidate the mechanisms and management of vaccination effects in children with PASC.

## Figures and Tables

**Figure 1 vaccines-12-00910-f001:**
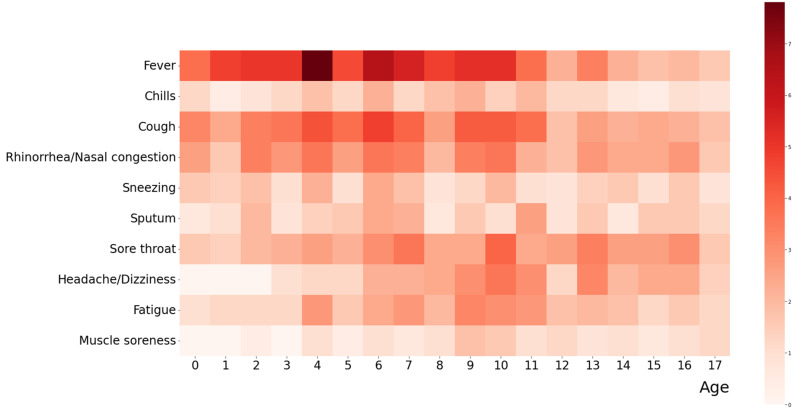
Distribution of top ten common clinical symptoms during the acute phase among 500 children with PASC, sorted by age. The heatmap illustrates the frequency of symptoms in children of different ages during the acute phase. Darker red shades represent higher frequencies. The color bar on the right indicates the number of occurrences, ranging from 0 (lightest) to 7 (darkest).

**Figure 2 vaccines-12-00910-f002:**
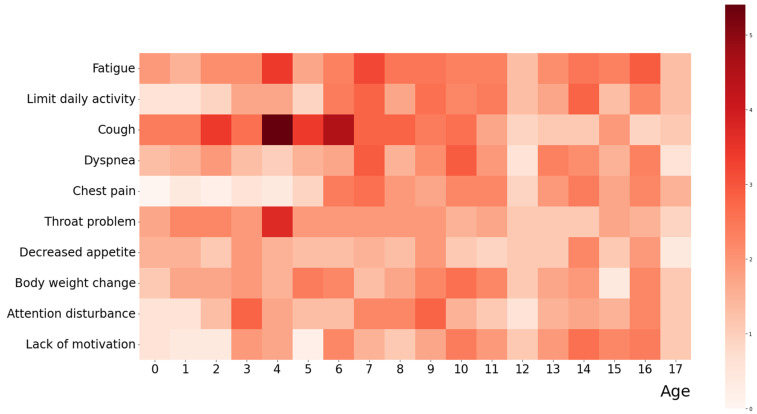
Distribution of top ten common clinical symptoms during the phase of the post-COVID-19 condition among 500 children with PASC, sorted by age. The heatmap displays the frequency of symptoms observed in children of different ages during the post-COVID-19 phase. Darker red shades correspond to higher frequencies. The color bar on the right represents the number of occurrences, with values ranging from 0 (lightest) to 5 (darkest).

**Figure 3 vaccines-12-00910-f003:**
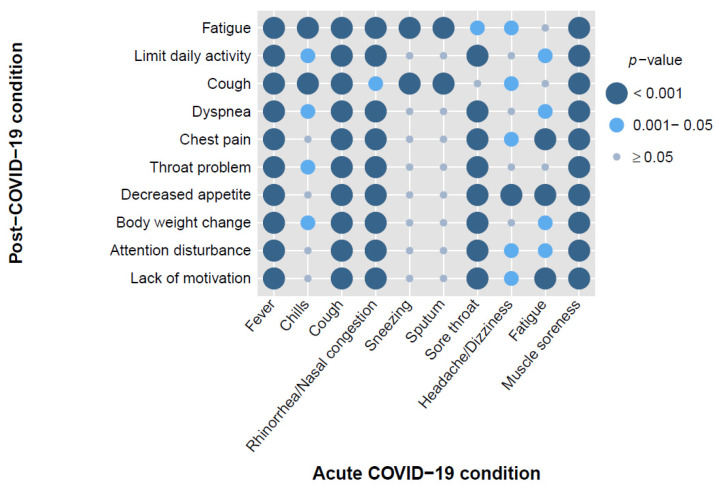
*P*-values distribution of clinical symptoms during the phases of acute and post-COVID-19 conditions for 500 children with PASC. The figure represents the significance of the coincidence between different clinical symptoms during the acute and post-COVID-19 phases. The size and color of the dots represent the *p*-value: dark blue dots (*p* < 0.001) indicate a highly significant coincidence, medium blue dots (*p* = 0.001–0.05) indicate a moderate coincidence, and light blue dots (*p* ≥ 0.05) indicate a non-significant coincidence.

**Figure 4 vaccines-12-00910-f004:**
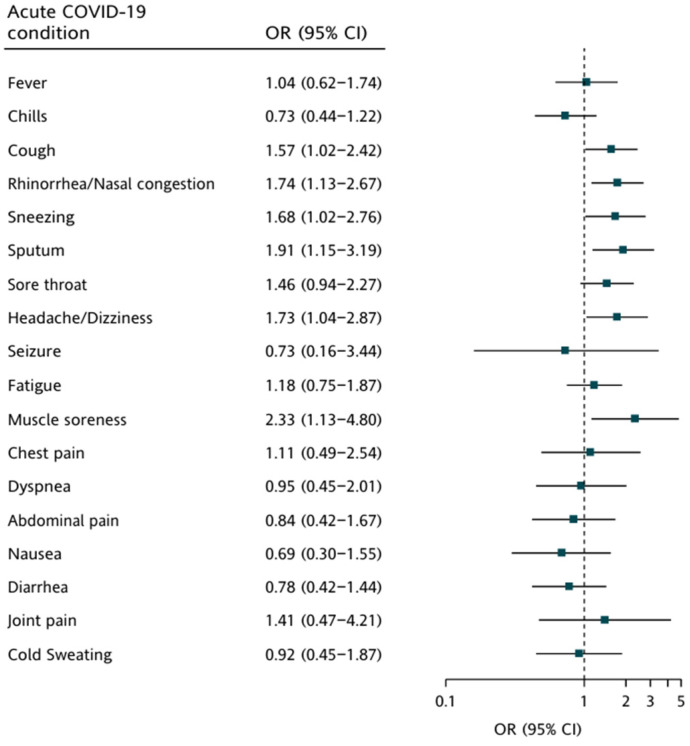
Association between clinical symptoms during the acute phase and vaccination in 500 children with PASC.

**Figure 5 vaccines-12-00910-f005:**
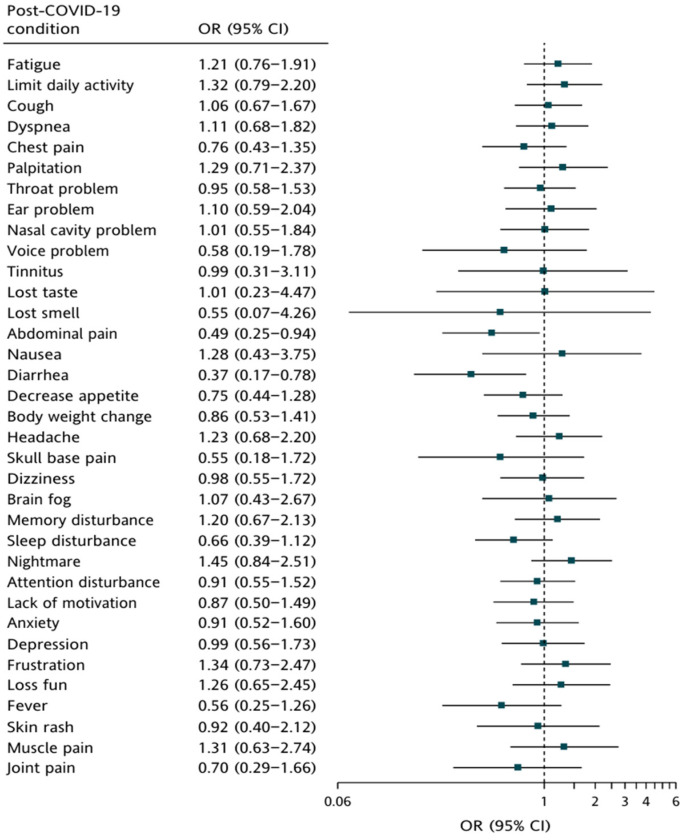
Association between clinical symptoms during the post-COVID-19 phase and vaccination in 500 children with PASC.

**Table 1 vaccines-12-00910-t001:** Baseline characteristics among 500 children with PASC, grouped by vaccination status.

	Overall (*N* = 500)	Vaccinated (*N* = 292)	Unvaccinated (*N* = 208)	*p* *
Demographic characteristics				
Age, years				
mean (SD)	7.58 (4.60)	9.69 (4.09)	4.62 (3.53)	**<0.0001**
median (IQR)	7 (4, 11)	9 (6.5, 13)	4 (2, 7)	
Sex, *n* (%)				
Female	218 (43.6)	137 (46.9)	81 (38.9)	0.08
Male	282 (56.4)	155 (53.1)	127 (61.1)	
Confirmed test, *n* (%)				
RT-PCR	134 (58.8)	57 (47.9)	77 (70.6)	**<0.0001**
Rapid antigen test	400 (88.1)	243 (89.0)	157 (86.7)	0.46
Hospitalization, *n* (%)				
Ward	35 (7.1)	16 (5.5)	19 (9.4)	0.10
ICU	2 (0.5)	0 (0.0)	2 (1.2)	0.10
Biomedical tests				
	*N*; mean ± SD	*N*; mean ± SD	*N*; mean ± SD	*p*
WBC (per μL)	309; 8.30 *±* 7.57	195; 8.21 ± 9.30	114; 8.45 ± 2.75	0.75
RBC (per μL)	309; 6.34 ± 27.66	195; 7.29 ± 34.82	114; 4.73 ± 0.43	0.31
Neutrophils (%)	359; 47.56 ± 13.00	227; 50.33 ± 11.24	132; 42.78 ± 14.40	**<0.0001**
Lymphocytes (%)	359; 40.75 ± 13.10	227; 38.11 ± 10.98	132; 45.31 ± 15.10	**<0.0001**
N/L ratio	1.17	1.32	0.94	
Monocytes (%)	359; 7.32 ± 2.90	227; 7.34 ± 3.11	132; 7.29 ± 2.52	0.87
Eosinophils (%)	359; 3.51 ± 3.02	227; 3.46 ± 2.98	132; 3.59 ± 3.10	0.71
Hemoglobin (g/dL)	308; 12.95 ± 1.49	194; 13.09 ± 1.48	114; 12.72 ± 1.47	**0.04**
Platelets (×10^3^ per μL)	124; 322.56 ± 93.70	94; 301.95 ± 71.53	30; 387.13 ± 123.06	**0.001**
ESR (mm/h) (normal range 0–20 mm/h)	171; 7.08 ± 6.55	94; 6.81 ± 7.10	77; 7.40 ± 5.84	0.56
hsCRP (mg/dL) (normal range 0.04–1.12 mg/dL)	353; 0.19 ± 0.62	225; 0.19 ± 0.65	128; 0.20 ± 0.57	0.81
LDH (U/L) (normal range 120–330 U/L)	312; 201.21 ± 57.47	199; 188.90 ± 49.53	113; 222.88 ± 63.97	**<0.0001**
Ferritin (ng/mL) (normal range 10–300 ng/mL)	297; 52.80 ± 43.14	191; 56.04 ± 40.17	106; 46.97 ± 47.67	0.10
D-Dimer (ng/mL) (normal range ≤ 500 ng/mL)	347; 312.24 ± 185.50	221; 290.78 ± 156.03	126; 349.88 ± 223.95	**0.01**
AST (U/L) (normal range 15–45 U/L)	322; 24.12 ± 10.65	207; 22.29 ± 10.35	115; 27.41 ± 10.42	**<0.0001**
ALT (U/L) (normal range 5–45 U/L)	323; 15.92 ± 19.48	207; 16.29 ± 21.12	116; 15.27 ± 16.21	0.63
CPK (U/L) (normal range 5–200 U/L)	311; 117.62 ± 98.16	200; 115.01 ± 104.88	111; 122.33 ± 84.98	0.50
Total IgE (IU/mL) (normal range < 100 IU/mL)	269; 315.46 ± 562.40	178; 358.19 ± 640.76	91; 231.88 ± 351.76	**0.04**

Abbreviations: PASC: post-acute sequelae of SARS-CoV-2 infection; SD: standard deviation; IQR: interquartile range; RT-PCR: reverse transcriptase-polymerase chain reaction; ICU: intensive care unit; WBC: white blood cell; RBC: red blood cell; N/L ratio: neutrophil/lymphocyte ratio; ESR: erythrocyte sedimentation rate; hsCRP: high-sensitive C-reactive protein; LDH: lactic dehydrogenase; AST: aspartate aminotransferase; ALT: alanine transaminase; CPK: creatine phosphokinase; IgE: immunoglobulin E. * *p* values less than 0.05 are shown in bold.

**Table 2 vaccines-12-00910-t002:** Baseline characteristics among 500 children with PASC, grouped by age.

	Total (*N* = 500)	0–5 Years (*N* = 180)	6–11 Years (*N* = 213)	12–17 Years (*N* = 107)	*p **
Demographic characteristics					
Age, years					
Mean (SD)	7.58 (4.603)	2.76 (1.659)	8.24 (1.706)	14.36 (1.621)	**<0.0001**
Median (IQR)	7 (4, 11)	3 (1, 4)	8 (7, 10)	14 (13, 16)	**<0.0001**
Sex, *n* (%)					
Female	218 (43.6)	77 (42.8)	88 (41.3)	53 (49.5)	0.36
Male	282 (56.4)	103 (57.2)	125 (58.7)	54 (50.5)	
Confirmed test, *n* (%)					
RT-PCR	134 (58.8)	62 (70.5)	45 (49.5)	27 (55.1)	**0.01**
Rapid antigen test	400 (88.1)	146 (87.4)	167 (87.4)	87 (90.6)	0.69
Hospitalization, *n* (%)					
Ward	35 (7.1)	18 (10.1)	12 (5.8)	5 (4.8)	0.15
ICU	2 (0.5)	1 (0.7)	1 (0.6)	0 (0.0)	0.76
COVID-19 vaccine, *n* (%)					
0	208 (41.6)	140 (77.8)	57 (26.8)	11 (10.3)	**<0.0001**
1	139 (27.8)	33 (18.3)	93 (43.7)	13 (12.1)	
2	95 (19.0)	6 (3.3)	55 (25.8)	34 (31.8)	
3	54 (10.8)	1 (0.6)	8 (3.8)	45 (42.1)	
4	4 (0.8)	0 (0.0)	0 (0.0)	4 (3.7)	
Biomedical tests					
	*N*; mean ± SD	*N;* mean ± SD	*N*; mean ± SD	*N*; mean ± SD	*p*
WBC (per μL)	309; 8.30 ± 7.57	92; 9.67 ± 9.60	144; 7.46 ± 2.61	73; 8.24 ± 10.54	0.09
RBC (per μL)	309; 6.34 ± 27.66	92; 9.97 ± 50.70	144; 4.80 ± 0.52	73; 4.82 ± 0.53	0.33
Neutrophils (%)	359; 47.56 ± 13.00	107; 40.12 ± 15.22	166; 47.93 ± 10.2)	86; 56.08 ± 8.73	**<0.0001**
Lymphocytes (%)	359; 40.75 ± 13.10	107; 47.42 ± 16.19	166; 40.61 ± 9.91	86; 32.75 ± 9.18	**<0.0001**
N/L ratio	1.17	0.85	1.18	1.71	
Monocytes (%)	359; 7.32 ± 2.90	107; 7.33 ± 2.75	166; 7.11 ± 3.31	86; 7.73 ± 2.11	0.27
Eosinophils (%)	359; 3.51 ± 3.02	107; 3.44 ± 3.25	166; 3.94 ± 3.20	86; 2.77 ± 2.08	**0.01**
Hemoglobin (g/dL)	308; 12.95 ± 1.49	92; 12.38 ± 1.78	143; 13.02 ± 1.19	73; 13.54 ± 1.37	**<0.0001**
Platelets (×10^3^ per μL)	124; 322.56 ± 93.70	23; 387.13 ± 124.91	58; 322.02 ± 85.07	43; 288.74 ± 65.31	**0.0002**
ESR (mm/h) (normal range 0–20 mm/h)	171; 7.08 ± 6.55	65; 8.05 ± 7.23	76; 6.99 ± 6.80	30; 5.20 ± 3.23	0.14
hsCRP (mg/dL) (normal range 0.04–1.12 mg/dL)	353; 0.19 ± 0.62	107; 0.27 ± 0.81	160; 0.15 ± 0.49	86; 0.19 ± 0.56	0.31
LDH (U/L) (normal range 120–330 U/L)	312; 201.21 ± 57.47	95; 242.97 ± 67.13	142; 198.73 ± 35.52	75; 153.00 ± 33.90	**<0.0001**
Ferritin (ng/mL) (normal range 10–300 ng/mL)	297; 52.80 ± 43.14	89; 43.74 ± 48.43	132; 51.81 ± 29.01	76; 65.15 ± 53.55	**0.01**
D-Dimer (ng/mL) (normal range ≤ 500 ng/mL)	347; 312.24 ± 185.50	105; 383.49 ± 251.73	157; 270.18 ± 101.43	85; 301.90 ± 185.49	**<0.0001**
AST (U/L) (normal range 15–45 U/L)	322; 24.12 ± 10.65	98; 30.09 ± 11.60	146; 23.50 ± 7.89	78; 17.77 ± 9.95	**<0.0001**
ALT (U/L) (normal range 5–45 U/L)	323; 15.92 ± 19.48	99; 15.16 ± 18.65	146; 15.49 ± 15.61	78; 17.71 ± 26.09	0.65
CPK (U/L) (normal range 5–200 U/L)	311; 117.62 ± 98.16	94; 128.71 ± 88.75	141; 117.75 ± 85.27	76; 103.66 ± 127.04	0.26
Total IgE (IU/mL) (normal range < 100 IU/mL)	269; 315.46 ± 562.40	79; 291.32 ± 478.39	117; 321.96 ± 658.14	73; 331.18 ± 480.21	0.90

Abbreviations: PASC: post-acute sequelae of SARS-CoV-2 infection; SD: standard deviation; IQR: interquartile range; RT-PCR: reverse transcriptase-polymerase chain reaction; ICU: intensive care unit; WBC: white blood cell; RBC: red blood cell; N/L ratio: neutrophil/lymphocyte ratio; ESR: erythrocyte sedimentation rate; hsCRP: high-sensitive C-reactive protein; LDH: lactic dehydrogenase; AST: aspartate aminotransferase; ALT: alanine transaminase; CPK: creatine phosphokinase; IgE: immunoglobulin E. * *p* values less than 0.05 are shown in bold.

## Data Availability

The data presented in this study are not publicly available due to privacy or ethical restrictions. Requests to access these datasets should be directed to the corresponding author.

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
