# Peer review of "Clinical Features and Vaccination Effects among Children with Post-Acute Sequelae of COVID-19 in Taiwan"

_vaccines, 2024, doi:10.3390/vaccines12080910_

Round 1
Reviewer 1 Report
Comments and Suggestions for Authors
The abstract is unclear.
The chosen limit of 4 weeks to classify patients as PASC is too short- It is already well known that the majority of pediatric patients spontaneously improve from persistent symptoms in some weeks, so to draw clinically significant conclusions and of interest in healthcare practice, it is much more appropriate to set a limit of 12 weeks.
The range of values ​​within normality in your Laboratory of analytical parameters is not indicated, so conclusions cannot be drawn in some cases- on the other hand, what cannot be considered relevant nor can values ​​within normality be classified as significant, such as, for example, with LDH or AST.
Figures 1 and 2 are poorly understood as the meaning of the colors of the caudrates is not well explained. Figure 3 is also not clear and not easy to understand. In both cases it would be convenient to use classic figures or tabals with direct and simple interpretation.
Comments on the Quality of English LanguageI have no specific comments
Author Response
Dear Editor and Reviewers:
Thank you very much for your insightful comments and extensive suggestions. All your comments we received on this study have been taken into account to improve our manuscript. We have completed the revision based on the Reviewers’ comments point by point in the table below. The changes were with a gray background, bold font, and underlining. We hope that these changes to the manuscript will facilitate the decision to publish this study in your journal.
We look forward to hearing good news from you.
Sincerely,
Corresponding author:
Hui-Ju Tsai, MPD, PhD,
Institute of Population Health Sciences, National Health Research Institutes, Zhunan, Taiwan, Department of Medical Science, National Tsing-Hua University, Hsinchu, Taiwan,
Email: tsaihj@nhri.edu.tw
Address: No. 35, Keyan Road, Zhunan Town, Miaoli County 350, Taiwan
Telephone number: +886-37-206166-36150
Fax number: +886-37-586467
Jiu-Yao Wang, MD, DPhil.,
Center of Allergy, Immunology and Microbiome (A.I.M), China Medical University Hospital and Children's Hospital, Taichung, Taiwan,
Email: a122@mail.ncku.edu.tw; wangjy@mail.cmu.edu.tw
Address: No. 2, Yude Rd., North Dist., Taichung City, 40447, Taiwan
Telephone number: +886-4-22052121-4698
Fax number: +886-4-22032798

Reviewer 2 Report
Comments and Suggestions for Authors
It's an important study for a better understanding of the pathomechanism of PASC in children. The authors have provided some details on the background, methods and techniques used to conduct the study. However, for better understanding, clarity and transparency, the authors are encouraged to make some additional changes before publication. Authors are encouraged to include the following suggestions in the manuscript:
90-91: Make the objective more precise and shorter by deleting the repetitions in line 90: ‘acute COVID-19 infection and PASC’ and line 91 ‘acute and post-COVID-19 phases, respectively’. Lines 88 and 89 already refer to the two phases.
95: The text states that this is a prospective cohort study, but does not mention which data collection points form the data basis. Is the analysis based on cross-sectional data? In the discussion part, the authors mention that the follow-up period was relatively short. How long then? It should be added to the methods section.
97-98: Instead of the term ‘study children’, use the term ‘study participants’. The above term is a bit bumpy, e.g. lines 97-98: The study participants were children from the DISCOVER study cohort (Diagnosis and Support for COVID Children to Enhance Recovery). Please adjust it in the complete manuscript.
99: multidisciplinary instead of multi-discipline
102: PASC instead of PSAC
107: What "relevant guidelines and regulations" are the study authors referring to? Please cite the regulations.
110: The description of the data collection does not mention that symptoms from the acute phase were also recorded
113: Were the study participants also asked how often they were vaccinated and when the last vaccination was administered? Because this plays an important role in the prevention of SARS-COV-2 infection. The longer the interval between the last vaccination and infection, the lower the level of protection.
115: Were blood samples taken from all study participants?
It is not yet clear to me what the authors' aim is in presenting the biomarkers. In my opinion, this needs to be made clearer in the background section and also in the objectives.
124-125: What goal are the authors pursuing with the age categorisation? Does age play a role in the development and severity of PASC? If so, please add reasons.
132-133/ Results: Do I understand correctly that the symptoms from the acute phase and post-acute phase were analysed with regard to vaccination status, i.e. graph 1 and graph S1 describe symptoms of vaccinated versus unvaccinated children. I would therefore clarify the statement: „Vaccinated children had a higher odds of reporting the following symptoms in the acute phase compared to unvaccinated children:..
140: avoid repetitions as far as possible: remove PASC:e.g. „A total of 500 PASC children with PASC were included in this study“
140-142: It should be clear by now that 500 children were included in the study. Avoid repetitions e.g. 500.
153: insted of „tended to have greater age“, replace with „tended to be older“
183: does the number of acute symptoms in vaccinated vs non-vaccinated children have an effect on the PASC development?
190: Figure S1 has to be presented in the manuscript as the results are also mentioned in the main text and the abstract
202-203 avoid repetitions: e.g. children / cohort
219: remove „, which“
222: The main results on age differences could have been presented in more detail (see line 160) and not just referred to additional material.
271-277: The authors provide no explanation as to why vaccination might have a protective effect on the gastrointestinal system in children. This is given far too little attention in the discussion. Perhaps it is also a coincidental finding?
Comments on the Quality of English Language
I would recommend having a native speaker check the manuscript for minor style and grammar issues.
Author Response

(The authors gave the same response as above.)

Reviewer 3 Report
Comments and Suggestions for Authors
The manuscript by Hsu et al tried to evaluate the clinical features and vaccination effects among children with post-acute sequalae of COVID-19 in Taiwan. My concerns are as follow:
1. How was the sample size of 500 PASC children determined? Could the authors provide evidence to support the idea that this sample size is enough for this investigation?
2. The authors declared that cough and throat problem were among the top ten common clinical symptoms for the post-COVID-19 condition. I am wondering what is the causal factor for these symptoms. If they were caused by SARS-CoV-2 infection, thus, the definition for PASC by the authors is inappropriate. If they were caused by some other infections, the authors should evaluate the influence, or exclude the children likely undergoing acute infections.
3. The author found significant positive associations of clinical symptoms in acute phase with vaccination. This conclusion seems irrational and difficult to understand. Because vaccination usually provides protection. Could the authors make it clear? or make it sure that the symptoms were not caused by other ongoing infection?
Comments on the Quality of English Language
Language polishing is recommended.
Author Response

(The authors gave the same response as above.)

Reviewer 4 Report
Comments and Suggestions for Authors
Long COVID in children is a real and concerning condition where symptoms persist long after the initial COVID-19 infection has cleared. The present manuscript describes a nicely designed and conducted study investigating the clinical features and vaccination effects in children with PASC. A well-written manuscript, with an adequate presentation of the results and discussion, including the positive aspects and limitations of the study. I agree with author’s conclusion about the value of the study in terms of providing a comprehensive investigation of baseline characteristics, and clinical symptoms in both acute and post-COVID-19 phases, and links vaccination effects to both acute and post-COVID-19 phases. Authors have confronted their results with the literature reports even with the understanding that the wide range of symptoms and their varying degrees of severity make it difficult to establish a standard definition and diagnostic criteria for PASC.
However, in a large-scale, exploratory study, (DOI: 10.1001/jamapediatrics.2022.2800), ref 4 of the manuscript, Dr Rao and colleagues reported that myocarditis was the most commonly diagnosed PASC-associated condition (it was a study with a cohort of 650’000 children. The adjusted hazard ratios (aHRs) was above 3. Also, in a recent review ( DOI: 10.3390/ijms24021147 ) on the COVID-19 Heart Lesions in Children: Clinical, Diagnostic and Immunological Changes, the authors consider that further studies are needed to evaluate whether transient or persistent cardiac complications are associated with long-term adverse cardiac events.
In addition, the Centers for Disease Control and Prevention (CDC), myocarditis has been identified as side effect of the mRNA vaccine (https://www.cdc.gov/coronavirus/2019-ncov/vaccines/safety/myocarditis.html). This side effect is seen much more often in male teens and young adults, as compared to other ages, and is likely due to an exaggerated immune response. It will be important to know –considering Dr. Rao study and the CDC statement about myocarditis- your comments. Myocarditis related variables may be represented by fatigue, chest pain, palpitations, or difficulty breathing (could suggest myocarditis) it would be nice to get authors discussion in the text about this relevant event even when the limitations of the study were well described.
Author Response

(The authors gave the same response as above.)

Round 2
Reviewer 1 Report
Comments and Suggestions for Authors
The article has improved compared to the previous version
Author Response
Dear Reviewer,
Thank you for taking the time to review this manuscript. All your comments we received on this study have been taken into account to improve our manuscript. We have completed the revision based on the Reviewers’ comments point by point as follows.
Comments 1: The article has improved compared to the previous version
Response 1: Thank you for your positive feedback and for acknowledging the improvements in our revised article. We appreciate your previous constructive comments, which have significantly contributed to enhancing the quality of our manuscript.
We look forward to hearing good news from you.
Sincerely,
Corresponding author:
Hui-Ju Tsai, MPD, PhD,
Institute of Population Health Sciences, National Health Research Institutes, Zhunan, Taiwan, Department of Medical Science, National Tsing-Hua University, Hsinchu, Taiwan,
Email: tsaihj@nhri.edu.tw
Address: No. 35, Keyan Road, Zhunan Town, Miaoli County 350, Taiwan
Telephone number: +886-37-206166-36150
Fax number: +886-37-586467
Jiu-Yao Wang, MD, DPhil.,
Center of Allergy, Immunology and Microbiome (A.I.M), China Medical University Hospital and Children's Hospital, Taichung, Taiwan,
Email: a122@mail.ncku.edu.tw; wangjy@mail.cmu.edu.tw
Address: No. 2, Yude Rd., North Dist., Taichung City, 40447, Taiwan
Telephone number: +886-4-22052121-4698
Fax number: +886-4-22032798
Reviewer 3 Report
Comments and Suggestions for Authors
I donot think the authors had fit my concerns.
1. For the sample size evaluation, although we lack the prevalence data of PASC in children in Taiwan, we could make full use of the related data from other area, for example the asia-pacific area. My suggestion is to evaluate the sample size from a statistical point of view, to determine if the enrolled 500 PASC is enough for this investigation.
2. For the definition of long covid of post-covid sympotoms, an important qualifer is the sympotoms could not be explained by any other causes. Although cough and sore throat could be included in the post-covid symptoms, the authors should declare that the cough symptom could not be explained by any other causes.
Comments on the Quality of English Language
none.
Author Response
Dear Reviewer,
Thank you for your detailed feedback and for bringing up these important points. We have carefully considered your comments and made the following revisions to address your concerns:
Comment 1: For the sample size evaluation, although we lack the prevalence data of PASC in children in Taiwan, we could make full use of the related data from other area, for example the asia-pacific area. My suggestion is to evaluate the sample size from a statistical point of view, to determine if the enrolled 500 PASC is enough for this investigation.
Response 1:
Thank you for this suggestion. We used an alpha level of 0.05 and a beta level of 0.20, with the proportion of subjects in the exposed group being 0.58 (based on our study of children with vaccination), a risk in the non-exposed group (baseline risk) of 0.25 (according to previous results from a meta-analysis, reference 6), and an odds ratio (OR) of 1.8. Using these parameters, we estimated the required sample size. The calculated sample size is approximately 443 subjects without continuity correction and 476 subjects with continuity correction.
Therefore, we think that our enrolled sample size of 500 PASC could be sufficient for this investigation.
Comment 2: For the definition of long covid of post-covid sympotoms, an important qualifer is the sympotoms could not be explained by any other causes. Although cough and sore throat could be included in the post-covid symptoms, the authors should declare that the cough symptom could not be explained by any other causes.
Response 2:
Authors totally agree with the Reviewer’s opinion that symptoms of Post-COVID, although not yet standardized, should not be explainable by any other cause. Although cough and sore throat are common symptoms, they last longer (over 4 weeks) than other common colds and have been ruled out for other maladies by our pediatricians. Hence, we included cough and sore throat as two unique characteristics of post-COVID symptoms that could not be explained by any other cause.
We have added the following sentences to the methods section for clarity: "Participants in this study were specifically evaluated in the PASC outpatient department to ensure they were experiencing post-COVID-19 conditions and not undergoing any acute infections. This evaluation included an assessment by a pediatrician to exclude any other acute infections or other causes that mimic post-COVID symptoms." in Page 3 Line 104-108.
We hope these revisions meet your expectations and address your concerns comprehensively. Thank you again for your constructive comments, which have significantly improved the quality of our manuscript.
We look forward to your positive feedback.
Sincerely,
Corresponding author:
Hui-Ju Tsai, MPD, PhD
Institute of Population Health Sciences, National Health Research Institutes, Zhunan, Taiwan
Department of Medical Science, National Tsing-Hua University, Hsinchu, Taiwan
Email: tsaihj@nhri.edu.tw
Address: No. 35, Keyan Road, Zhunan Town, Miaoli County 350, Taiwan
Telephone number: +886-37-206166-36150
Fax number: +886-37-586467
Jiu-Yao Wang, MD, DPhil
Center of Allergy, Immunology and Microbiome (A.I.M), China Medical University Hospital and Children's Hospital, Taichung, Taiwan
Email: a122@mail.ncku.edu.tw; wangjy@mail.cmu.edu.tw
Address: No. 2, Yude Rd., North Dist., Taichung City, 40447, Taiwan
Telephone number: +886-4-22052121-4698
Fax number: +886-4-22032798